# Electric current control of spin helicity in an itinerant helimagnet

N. Jiang [1✉], Y. Nii[2], H. Arisawa[2], E. Saitoh[2,3,4,5] & Y. Onose[2✉]

A helimagnet is a chiral magnet in which the direction of the magnetic moment spatially rotates in a plane perpendicular to the propagation vector. The sense of the rotation known as spin helicity is a robust degree of freedom of matter and may provide a new concept of magnetic memory if it can be electrically controlled and detected. Here we show that the helicity can be controlled by magnetic fields and electric currents in an itinerant helimagnet MnP. Second-harmonic resistivity measurements allow us to read out the controlled helicity. In contract to an insulating multiferroic magnet, in which spin rotation was shown to be controllable by an electric field, we achieve helicity manipulation by using an electric current in the conducting helimagnet. The controllability of the spin helicity may pave the way to new method of realizing magnetic memories based on the spin internal degrees of freedom.

[1] Department of Basic Science, The University of Tokyo, Tokyo 153-8902, Japan. [2] Institute for Materials Research, Tohoku University, Sendai 980-8577, Japan. [3] Department of Applied Physics, The University of Tokyo, Tokyo 113-8656, Japan. [4] Advanced Science Research Center, Japan Atomic Energy Agency, Tokai 319-1195, Japan. [5] Advanced Institute for Materials Research, Tohoku University, Sendai 980-8577, Japan. ✉email: jiangnan@g.ecc.u-tokyo.ac.jp; onose@imr.tohoku.ac.jp

Chirality is the breaking down of mirror symmetry in matter. In the fields of biology and chemistry, it is particularly important because some of the essential molecules in human bodies, such as amino acids and DNA, have a chirality. It is a long-standing mystery how one of the enantiomers was chosen at the beginning stage of life on earth[1,2]. Recently, chirality has been attracting attention in magnetism and materials science, and some spiral magnetic structures show chirality. There are two types of spiral magnetic structures. One of them is a helical structure (longitudinal spiral or proper screw structure) in which the ordered direction of the magnetic moment spatially rotates in a plane perpendicular to the propagation vector (Fig. 1a), whereas for the other type of spiral magnet known as a transverse spiral magnet or cycloid structure, the spiral plane is parallel to the propagation vector. The helical structure hosts the chirality because any mirror operation reverses the sense of rotation, referred to as helicity. The helicity degree of freedom is robust against disturbances, and is therefore useful for retaining information. Recently, spintronics based on antiferromagnets has attracted much attention because of the high-speed dynamics and robustness against stray fields[3,4]. The helical magnetic structure is one form of antiferromagnetic structure and seems to share the benefits mentioned above. Realization of control and detection of helicity may open up new possibilities for magnetic memory applications based on helimagnets.

There are several origins of spiral magnetic structures. One of them is the noncentrosymmetric crystal structure. The breaking of the space-inversion symmetry of the crystal structure induces a long-period spiral spin structure, owing to the antisymmetric magnetic interaction known as the Dzyaloshinskii–Moriya (DM) interaction[5,6]. In this case, the spin-rotation direction is locked to the noncentrosymmetric crystal structure, and external fields cannot reverse it. Another origin of the spiral magnetic structure is magnetic frustration; competing magnetic interactions may give rise to spiral magnetic structures[7]. In itinerant magnets, the magnetic interactions mediated by conduction electrons, such as the Ruderman–Kittel–Kasuya–Yoshida interaction and nesting effects of the Fermi surface, also induce the spiral magnetic structure[8,9]. Except for the first case, the spiral magnetic transition involves the breaking of the space-inversion symmetry, and the spin-rotation direction serves as an internal degree of freedom.

It is of great interest how the spin-rotation direction couples with electromagnetic fields. In insulators, the spiral magnetic structure often induces ferroelectric polarization, as studied in the field of multiferroics[10–12]. The sign of the polarization is determined by the sense of the spin rotation, which can be controlled by an electric field. The magnitude of the ferroelectric polarization depends on the angle between the spiral plane and the propagation vector; the polarization shows a maximum when the propagation vector is parallel to the spiral plane (cycloid structure) and vanishes for the perpendicular configuration (helical structure). These are related to the magnetic symmetry. In cycloid structure, the spin-rotation direction is unchanged by a mirror operation parallel to the spiral plane and by that perpendicular to the propagation vector, but it is reversed by a mirror operation perpendicular to both the former two mirrors, being consistent with polar symmetry (see Supplementary Note 1). Note that the magnetic moment is an axial vector, and therefore it is reversed by a mirror operation parallel to it, but not by a perpendicular mirror operation. In helical structure on the other hand, the helicity is reversed by any mirror operation, which indicates chiral symmetry.

Coupling between the spin-rotation direction and electric current in itinerant spiral magnets is almost unknown. It is worth noting that an electric field is screened and vanishes in metallic media. To elucidate this issue, let us discuss the effect of spin-transfer torque in spiral magnets. For smooth spin textures, the spin-transfer torque under the electric current **j** can be expressed as:

$$\boldsymbol{\tau}_{\mathrm{STT}} = A(\mathbf{j} \cdot \nabla)\mathbf{S} + \beta \mathbf{S} \times (\mathbf{j} \cdot \nabla)\mathbf{S}, \quad (1)$$

where **S** is position-dependent magnetization vector, and $A$ and $\beta$ are constants[13,14]. The first term is adiabatic spin-transfer torque,

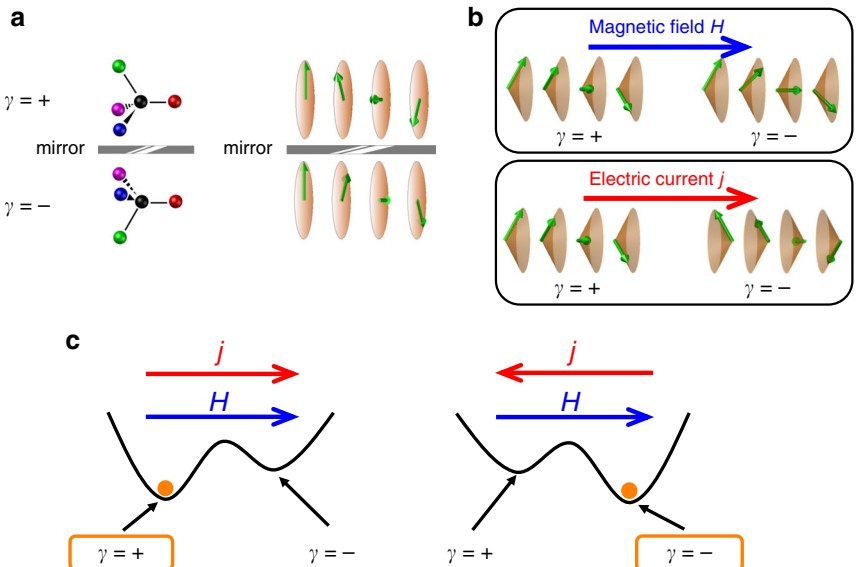

**Fig. 1 Electric current control of helicity in an itinerant longitudinal spiral magnet. a** Illustrations of chiralities of molecules and spiral magnets. $\gamma = \pm$ denotes the chirality (helicity). **b** Illustrations of the magnetic structure of spiral magnets in the presence of a magnetic field or electric current. In the presence of a magnetic field, the magnetic moments are tilted to the field direction, forming a conical magnetic structure. The net magnetization is along the magnetic field, irrespective of the chirality. The electric current along the propagation vector also induces the conical magnetic structure. In this case, the net magnetization direction is dependent on the chirality. **c** Illustrations of the electric current control of helicity in a longitudinal spiral magnet. The energetically favored helicity in the presence of magnetic field $H$ and electric current $j$ depends on whether they are parallel or antiparallel to each other.

and the second term is the so-called $\beta$-term, which originates from the spin relaxation and the non-adiabaticity. When an electric current is applied along the propagation vector of a helical magnet, the first term rotates the spin spiral structure around the propagation vector, and the second term produces a cone-like (conical) spin structure[15]. The net magnetization direction of the conical spin structure depends on the sign of the product of spin helicity and electric current (Fig. 1b). The helicity dependence is in contrast with the effect of a magnetic field; the magnetization direction is independent of the helicity. One can extrapolate from them that the spin helicity degeneracy is lifted under a magnetic field and an electric current (Fig. 1c). In other words, the simultaneous application of the magnetic field $H$ and the electric current $j$ is expected to fix the spin helicity.

By means of this mechanism, we here demonstrate helicity control with the use of magnetic fields and electric currents for an itinerant helical magnet MnP. This is totally distinct from the inverse DM mechanism for multiferroic spiral magnets. The controlled helicity is detected by second-harmonic resistivity. The successes of control and detection of helicity degree of freedom may contribute to future magnetic memory application.

## Results

**Properties of the MnP sample**. MnP has an orthorhombic and centrosymmetric crystal structure with the space group *Pbnm* (Fig. 2a)[16–23]. To increase the electric current density, we fabricated a micrometer-scale single-crystalline sample ($\sim 10 \times 20 \times 1\ \mu m^3$) by using a focused ion-beam technique (see Methods and Supplementary Fig. 1). Figure 2b shows the resistivity $\rho$ for the present microfabricated sample compared with that for a bulk (millimeter scale) sample previously reported by Shiomi et al.[21]. The resistivities are similar to each other, indicating that sample damage due to the microfabrication is minimal for the present sample. Figure 2c shows the magnetic-phase diagram constructed based on the magnetic field dependence of electrical resistivity for the present sample (see Supplementary

Note 2). The phase diagram is quite similar to that reported in the literature[23]. The ferromagnetically ordered phase (FM1) is stabilized even at zero magnetic field above 60 K, in which the magnetic moments align along the $c$-axis. We observed discontinuous changes in resistivity accompanying the hysteresis (inset in Fig. 2b), indicating first-order phase transition to the spiral magnetic state. Whereas the lower boundary temperature of the metastable region is 43.5 K, the higher boundary is indefinite, between 55 K and 60 K. In the spiral magnetic phase, the spiral plane is normal to the propagation vector along the $a$-axis[19]. Therefore, it is the helical structure realized in a centrosymmetric material. When the magnetic field is applied parallel to the $a$-axis, the magnetic moments are tilted, forming a conical magnetic state (Fig. 2d). As the magnetic field increases further before the magnetic moments become completely aligned along the $a$-axis (FM2), there emerges a fan structure (FAN), in which the magnetic moments are within the $ac$ plane, and the angle between the magnetic moment and the propagation vector spatially oscillates along the propagation vector (Fig. 2d). This magnetic structure can be viewed as a superposition of two conical structures with different helicities. Therefore, the fan-conical magnetic transition corresponds to the achiral–chiral transition.

**Electric current control of helicity**. In the case of ferroelectrics, the so-called poling procedure is frequently used for controlling electric polarization. In this procedure, an electric field is applied in a high-temperature para-electric phase, and then the temperature is decreased below the ferroelectric transition temperature. One can easily control the polarization with the poling procedure owing to a small coercive field and large susceptibility around the transition temperature. Here we adopt a similar method to control the helicity. We applied a dc electric current $j_\mathrm{p}$ parallel or antiparallel to the magnetic field $H_\mathrm{p}$ along the propagation vector ($a$-axis) when the fan-conical transition field was traversed (Fig. 2c, d). The magnitude of $H_\mathrm{p}$ was slowly decreased from 7 T to 3 T at a rate of 4 Th$^{-1}$ at 51 K just below the

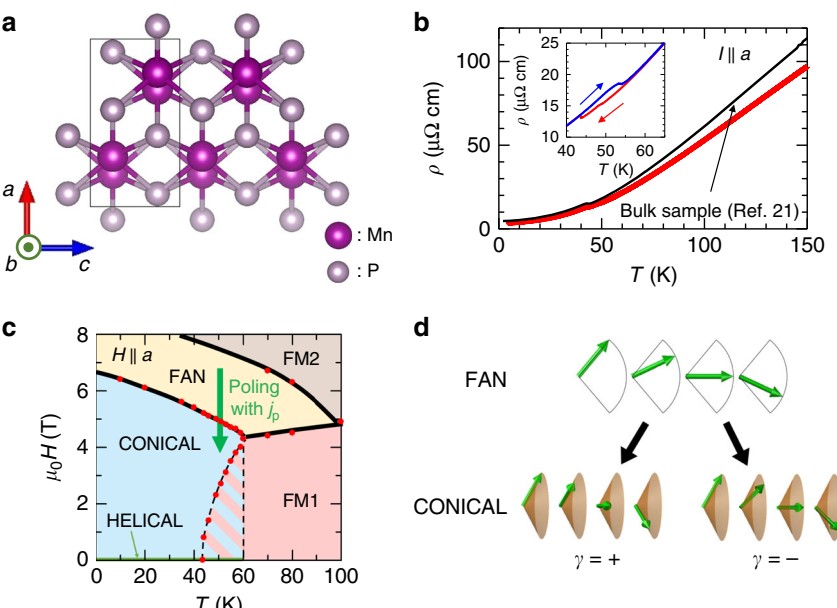

**Fig. 2 Properties of the MnP sample. a** Crystal structure of MnP. The black line represents the unit cell. **b** Temperature dependence of the resistivity for the present MnP sample fabricated by using a focused ion beam. The resistivity of a bulk (millimeter scale) sample previously measured by Shiomi et al.[21] is reproduced for comparison. The inset shows the history dependence of resistivity around the helical–ferromagnetic phase boundary. **c** The phase diagram of the microfabricated MnP sample for *H*||*a*. The red dots are the phase boundaries estimated by the magnetic field dependence of electrical resistivity (see Supplementary Note 2). The black lines are guides for the eyes. In the hatched region, the realized magnetic structure depends on the hysteresis. The green arrow suggests the poling procedure. **d** Illustrations of fan and conical magnetic structures.

ferromagnetic–helical transition temperature. Then, we removed the dc electric current $j_p$. Hereafter, we call the series of these processes a "poling procedure". To detect the helicity after the poling procedure, we utilize the electrical magnetochiral effect[24]: an electronic transport version of magnetochiral dichroism[25]. In the original magnetochiral phenomenon, the optical properties are dependent on the direction of the wave vector irrespective of the optical polarization. The effect shows up only when the time-reversal and mirror symmetries are simultaneously broken. Either chirality reversal or time reversal causes a sign change of the magnetochiral effect. In the electronic transport phenomenon, the resistivity $\rho$ is dependent on the current $j$ as follows:

$$\rho(j) = \rho_0 + \rho_{ch}j, \qquad (2)$$

where $\rho_0$ and $\rho_{ch}$ are constants. In this case, an ac electric current $j = j_{ac}\sin(\omega t)$ gives rise to the nonlinear voltage

$$V^{2nd}(t) = \rho_{ch}j_{ac}^2\cos(2\omega t). \qquad (3)$$

Here, $\omega$ and $t$ are angular frequency and time, respectively. While nonchiral noncentrosymmetric materials also show similar 2nd harmonic resistivities[26,27], a characteristic feature of the electrical magnetochiral effect is that the nonlinear transport is emergent parallel or antiparallel to the applied magnetic field. Such a phenomenon has been observed in several helimagnets with chiral crystal structures[28,29]. Since it has well been established that $\rho_{ch}$ depends on the chirality, the effect is useful as a prove for the sign of the chirality.

Figures 3a to d show the 2nd harmonic contribution of electrical resistivity $\rho^{2f}$ at 51 K as a function of the magnetic field along the propagation vector ($a$-axis), observed after the poling with $\pm j_p$ and $\pm H_p$. While the magnetic structure is history-dependent in the low-field range at 51 K, the helimagnetic state is stable in the field-decreasing process (see Supplementary Note 2). The magnitudes of poling and ac electric current densities $j_p$, $j_{ac}$ are $1.1 \times 10^9$ Am$^{-2}$ and $5.1 \times 10^8$ Am$^{-2}$, respectively. The observed magnetic field dependence of $\rho^{2f}$ may be viewed as a combination of odd and even functions. Whereas the even-function component seems to show small magnetic field dependence, the abrupt change around $H = 0$ indicates a notable odd-function component. The observation of the odd-function component is a hallmark of electrical magnetochiral phenomena specific to chiral symmetry, whereas the even-function contribution seems to stem from the inevitable effects of nonuniformity and/or electrode geometry, as discussed in the literature[26,29]. While some frequency dependence is discerned for the extrinsic even-function contribution, the odd-function contribution is almost independent of the frequency below 15 Hz (see Supplemental Note 3). $\rho_{ch}$ in Eq. (2) corresponds to the odd-function contribution. Since the crystal of MnP is centrosymmetric, the helical magnetic order is responsible for the emergence of the magnetochiral phenomenon. Importantly, the magnetochiral contribution shows a sign change by the reversal of either $H_p$ or $j_p$. These clearly show that the helicity of the helimagnetic structure depends on whether $H_p$ and $j_p$ are parallel or antiparallel.

**Variation of electrical magnetochiral effect.** To further examine the electrical magnetochiral effect and helicity control, we extract the intrinsic odd-function component of $\rho^{2f}$ by anti-symmetrization of the magnetic field dependence as $\rho_{asym}^{2f} = (\rho^{2f}(H) - \rho^{2f}(-H))/2$. As shown in Fig. 3e, $\rho_{asym}^{2f}$ at 0.4 T is proportional to $j_{ac}$, being consistent with the magnetochiral origin. Figure 3f shows $\rho_{asym}^{2f}$ at 51 K and 0.4 T as a function of $j_p$. $\rho_{asym}^{2f}$ is saturated above $j_p = 1 \times 10^9$ A/m$^2$, suggesting that the volume fraction of controlled helicity is nearly unity.

In Fig. 4a, b, we show the magnetic field dependence of $\rho_{asym}^{2f}$ below 3 T at various temperatures measured after the poling procedure at 51 K. With decreasing temperature from 51 K, $\rho_{asym}^{2f}$ steeply decreases and almost vanishes below 48 K. On the other hand, when the temperature increases from 51 K, $\rho_{asym}^{2f}$ rapidly increases up to 52 K, and then decreases with temperature. Figure 4c shows the temperature dependence of $\rho_{asym}^{2f}$ at 0.4 T. The data clearly show the sharp enhancement of $\rho_{asym}^{2f}$ at the phase boundary. In helimagnets with noncentrosymmetric crystal structures, the magnetochiral effect has been observed in a wider temperature range[28,29]. In MnSi, the magnetochiral signal shows an enhancement around the helical–paramagnetic transition temperature, but it is suppressed by applying a large magnetic field[28]. The origin of this was ascribed to the chiral magnetic

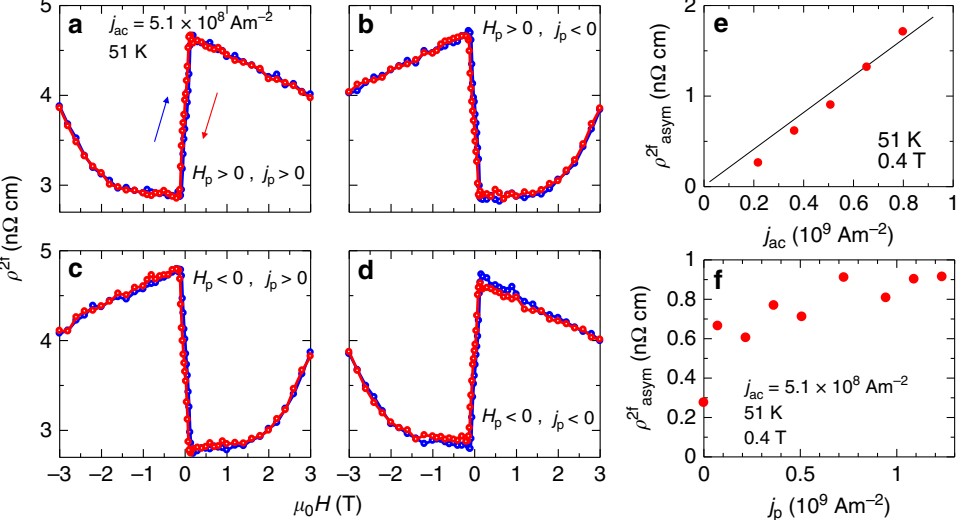

**Fig. 3 Electric current control of helicity. a–d** Magnetic field dependence of the 2nd harmonic contribution of electrical resistivity $\rho^{2f}$ after the poling procedure with the positive and negative magnetic fields $H_p$ and dc electric currents $j_p$. The magnitudes of $j_p$ and ac electric current for the measurement $j_{ac}$ are $1.1 \times 10^9$ Am$^{-2}$ and $5.1 \times 10^8$ Am$^{-2}$, respectively. **e** $\rho_{asym}^{2f}$ at 51 K and 0.4 T as a function of $j_{ac}$ measured after poling with $j_p = 1.1 \times 10^9$ Am$^{-2}$. **f** $\rho_{asym}^{2f}$ at 51 K and 0.4 T as a function of $j_p$.

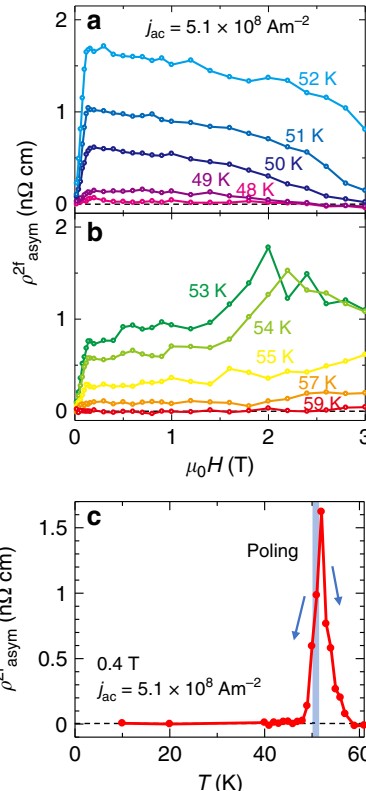

**Fig. 4 Temperature and magnetic field dependencies of magnetochiral effect. a, b** Magnetic field dependence of $\rho_{asym}^{2f}$ at various temperatures measured after the poling procedure at 51 K with $j_p = 1.1 \times 10^9\,\mathrm{Am^{-2}}$. **c** Temperature dependence of $\rho_{asym}^{2f}$ at 0.4 T.

fluctuation in the vicinity of the phase boundary. Enhancement of the magnetochiral signal around the phase boundary was also observed in $CrNb_3S_6$, while some components with different origins seem to coexist[29]. In this case, the magnetochiral signal increases more sharply as the ferromagnetic phase transition is approached from a lower temperature. Then, it rapidly decreases above 52 K owing to the reduction of helical volume fraction, because the inversion symmetry is unbroken in the ferromagnetic state. It should be noted that the magnetic field dependence of $\rho_{asym}^{2f}$ is contrastive with the linear magnetization curve along the *a*-axis[16]. Similar sharp enhancement at a low magnetic field unrelated to the magnetization is observed in the paramagnetic state of $MnSi$[28], which may be a characteristic of the chiral fluctuation-induced magnetochiral signal. Importantly, the strong chiral fluctuation in the vicinity of the phase boundary should make the magnetic structure electrically susceptible, which should more readily enable controllability of the spin helicity.

## Discussion

We have demonstrated control of the helicity (chirality) in a helical magnet MnP by using an electric current and a magnetic field, which satisfies the symmetrical rule that the external stimulus for chirality control should be collinear polar and axial vectorial fields with the same time-reversal symmetry (both odd or both even)[1,2]. Since control of a chiral object with a magnetic field and an electric or thermal current was not demonstrated previously, the applicability to a wide range of magnetic and chemical chiral objects should be examined. While antiferromagnetic spintronics has been attracting much attention recently, the present result opens up the possibility of magnetic memory based on helical magnets because the helicity degree of

freedom has now been shown to be controllable. Nevertheless, there are still a few issues to be resolved in future to realize such application. One of them is instantaneous helicity switching. While we achieved helicity control by using the poling procedure just below the transition temperature for the present microfabricated single crystal, practical devices need instantaneous helicity switching without any adjustment of magnetic field and temperature. In principle, an electric current with high enough density should be able to control the helicity instantaneously. Such a high-density electric current should be realized in the form of a thin-film sample. In addition, the voltage signal for a helicity probe becomes larger and more easily detectable for a thin-film sample. There is one report of thin-film fabrication of spiral magnets with a transition temperature above room temperature[30]. Another issue is the use of high magnetic fields. In this work, we applied a magnetic field to control and to detect helicity, but it cannot be used in practical electronic devices. Instead of applying a magnetic field, a spin-polarized current may be used for control and detection of the helicity. In fact, some theories suggest coupling between the helicity and spin current[31,32]. A junction with a spin-polarized ferromagnet or spin Hall material (e.g., Pt) seems useful for zero-field detection of the spin helicity.

## Methods

**Crystal growth**. We grew a single crystal of MnP, utilizing a Bridgman method. A silica tube with Mn and P was placed in a Bridgman furnace and slowly heated to 1200 °C within 14 days and kept at that temperature for 4 days. After that, the tube was moved toward the low-temperature region in a temperature gradient (~1–5 °C mm⁻¹) at a rate of 0.5 mm h⁻¹ for 21 days.

**Devise fabrication**. We extracted a microscale thin rectangular piece with an approximate size of $10 \times 20 \times 1\mu m^3$ from the crystal by using a focused ion-beam (FIB) technique (see Supplementary Fig. 1). The thin plate was mounted on a silicon wafer and fixed by FIB-assisted carbon deposition. Gold electrodes for four-probe resistivity measurements were fabricated using electron beam lithography and electron beam deposition.

**Resistivity measurement**. We measured the first and second-harmonic ac resistivities with an electric current frequency of 14.3 Hz in a superconducting magnet.

## Data availability

The source data underlying Figs. 2–4 and Supplementary Figs. 4, 5 are provided as a Source Data file. Data that support the findings of this study are available from the corresponding author upon reasonable request.

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

## Acknowledgements

The crystal growth was carried out by joint research in the Institute for Solid State Physics, The University of Tokyo with the help of R. Ishii and Z. Hiroi. The fabrication of the sample device was carried out partly by collaborative research in the Cooperative Research and Development Center for Advanced Materials, Institute of Materials Research, Tohoku University with the help of K. Takanashi and T. Seki. The authors thank G.E.W. Bauer, Y. Shimamoto, Y. Togawa, and J. Ohe for fruitful discussions. This work was supported in part by JSPS KAKENHI grant numbers JP16H04008, JP17H05176, JP18K13494, and JP19H05600 and the JST ERATO Spin Quantum Rectification Project (JPMJER1402). N.J. is supported by a JSPS fellowship (No. JP19J11151).

## Author contributions

N.J. carried out the crystal growth, device fabrication with the focused ion beam and electron beam lithography, and measurements of the magnetochiral effect. Y.N. contributed to the crystal growth and measurements of the magnetochiral effect. H.A. and E.S. contributed to the device fabrication with a focused ion beam. Y.O. conceived and supervised the project. N.J. and Y.O. wrote the paper with input from Y.N., H.A., and E.S.

## Competing interests

The authors declare no competing interests.
