## [Peer Review File · Nature Communications]

Reviewers' comments:

Reviewer #1 (Remarks to the Author):

Report by Lars Nordström, Uppsala University, Sweden on NCOMMS-19-32235.

This manuscript describes a novel way to write and read off the sign of the chirality of a magnetic helix by means of electrical currents. The idea and procedure is well described in a pedagogical way and the results are clear and convincing. This discovery has large potentials of impact in the field and is of interest for many other disciplines where chirality plays an important role.

Reviewer #2 (Remarks to the Author):

The Authors reports the control of chirality, originating from the helical spin order of MnP, by mutual application of magnetic field and electrical current. The presence of these two stimuli removes the degeneracy of the chiral enantiomers, being the right- and left-handed screw-like spin textures.

Using this principle to break both the inversion and the time reversal symmetry by combined magnetic field and electrical current, they manage to pole the system into a chiral (helical) mono-domain state.

They use the inverse effect, the so-called electrical magnetochiral effect (non-linear resistance dependent on the magnetic field and the current density) to detect the excess of the chiral enantiomer, favoured by poling.

I think these results are very interesting and novel in terms of fundamental understanding of control of helicity in itinerant magnets. Beyond this the Authors emphasize, and -in my opinion- overemphasize, the relevance to these results to spintronic applications. I would be more careful at this point for the following reasons:

i) they measure a non-linear (or second harmonic) resistance, coming from the non-reciprocal charge transport. To prove that this signal measures excess of a chiral enantiomer, one needs to demonstrate that the effect is odd in magnetic field as well as in current. This is nicely worked out in this work but this protocol is more complex than a simple second harmonic voltage detection.
ii) more importantly the effect is weak, meaning that the second harmonic resistance is on the level of nano-Ohm*cm for a chiral mono-domain sample exposed to magnetic fields in the Tesla range and current densities of $\sim 10^9 \text{ A/m}^2$. This leads to a voltage drop on the micro-Volt level over ~ 10 micron distances.

Therefore, if the Authors believe the effect explored here is relevant to applications, they should be more specific about the scheme of the device application and show its feasibility.

I found the main stream of the paper clear and likely interesting enough for publication in Nature Communications. However, before making a definite assessment I kindly ask the Authors to consider my comment above and a few technical ones below.

-The term helical is used in a confusing fashion. The helical spin structure means that the spins rotate in the plane normal to the magnetic modulation vector. The general term is magnetic spiral, with the helical and cycloidal structures being the limiting cases. The former is analogous to a structural helix, not the latter. I would suggest to use spiral for the general case and limit the usage of helical for the screw-like structure.

-In Fig. 1b the schematic representation of the mirror plane is misleading: For the case of the magnetic field the mirror plane should be perpendicular to the field. Basically, those states are

degenerate, which are connected by the symmetries of the fields. For the current these are mirror planes parallel to the current but for the magnetic field it is only the mirror plane perpendicular to the field. This may be better visualized in the figure.

In the main text they have the corresponding statement: "The net magnetization direction of the conical spin structure

depends on the sign of the product of spin helicity and electric current". Either a reference is needed or this should be explained based on the modified Fig. 1b or in analogy with a simple solenoid.

-The Authors should stress more that Eq. 1 is about a non-reciprocal term to the non-linear resistivity.

-It would be very interesting to see if the chiral domains can also be controlled in situ by current pulses applied in the chiral state and not only by poling through the magnetic transition. Have the Authors tried something along this line?

-What was the frequency range covered in the present study? Was there any frequency dependence of the effect observed?

-Phrasing and readability can generally be improved throughout the text.

Yours sincerely, Istvan Kezsmarki

Reviewer #3 (Remarks to the Author):

Dear Authors,

I do not recommend this paper for publication. The decision is mostly based on the language and on the unclear explanations used in the manuscript. Both points make it very hard to understand the text and to make a decision on the validity/novelty of the experiments or if it is of interest to the community.

The manuscript needs to be rewritten before being considered to be published anywhere. Here are some of the points:

a) The part of controlling the helicity of the spin spiral is not well explained. I think it is the poling process that induces a certain type of spin spiral, which they then detect by eMChE. However, due to bad explanations the reader cannot follow their logical process.

b) Very bad English: " To elucidate this issue, a theory paper regarding the spin dynamics in longitudinal helical magnets under electric current seems quite helpful." "Since the sense of rotation, which is denoted as helicity, is reversed by any mirror operation, it is corresponding to the chirality."

c) The whole explanation about the mirror planes is very poorly and unclearly explained. Figure 2 in the supplementary makes it even worse (what are x,y,z planes? It could be the planes where x,y,z are perpendicular to it, but this does not fit to the sketches.). Why is this even important for the paper? The paper does not discuss the transverse helical structure at all still it is extensively explained in the introduction.

d) Fig. 2c How is the black line between the FM1 and helical region (not the dotted line) determined? There are no experimental data points (red). Why does the phase diagram only go up to 100 K even though the text says that the magnetic phase starts at 290 K?

e) Fig. 2d why is there no sketch of FM1?

f) The eMChE method has been explained in other papers, however, a basic introduction to the main points are important to understand the results. This is the only explanation the reader gets: "While similar 2nd harmonic resistivities are also observed for non-chiral noncentrosymmetric materials, an important feature of electrical magnetochiral effect is that the nonlinear transport is

emergent in parallel or antiparallel with the applied magnetic field. This phenomenon has been observed in several chiral helimagnets. Since it has been well established that ρ depends on the chirality, it is useful for the probe."

g) What is a field-odd contribution?

h) Wrong use of articles throughout the whole manuscript.

Regards,

Authors' response to the referee 2's comments

First of all, we thank the referee for valuable comments.

Followings are our straightforward responses to them.

Comment(1)

Beyond this the Authors emphasize, and -in my opinion- overemphasize, the relevance to these results to spintronic applications. I would be more careful at this point for the following reasons:

i) they measure a non-linear (or second harmonic) resistance, coming from the non-reciprocal charge transport. To prove that this signal measures excess of a chiral enantiomer, one needs to demonstrate that the effect is odd in magnetic field as well as in current. This is nicely worked out in this work but this protocol is more complex than a simple second harmonic voltage detection.

ii) more importantly the effect is weak, meaning that the second harmonic resistance is on the level of nano-Ohm*cm for a chiral mono-domain sample exposed to magnetic fields in the Tesla range and current densities of $\sim 10^9 \text{A/m}^2$. This leads to a voltage drop on the micro-Volt level over ~ 10 micron distances.

Therefore, if the Authors believe the effect explored here is relevant to applications, they should be more specific about the scheme of the device application and show its feasibility.

Response(1):

We believe that it is important to suggest possible applications based on a new finding in a basic research paper. It sometimes leads to a real application after overcoming some technical issues. In this case, the helicity control is truly a new finding. We really believe that this may be a big breakthrough toward "helical spintronics".

As the reviewer suggested, the present method of chirality detection cannot be directly applied to the commercial device because of the smallness of the signal and the usage of the magnetic field. We believe we can overcome these technical issues with the use of thin-film and the spin current near future.

The mechanism we found should be common to other helimagnets. Actually, there is some helimagnet with the transition temperature above room temperature, of which the single-crystal thin film is previously reported (J. Magn. Magn. Mater. 421, 336 (2017)). The voltage signal should be enhanced for the thin film. For the detection of helicity at

zero magnetic fields, the spin-polarized current can be used instead of applying the positive and negative magnetic fields. In fact, some theories suggest the coupling between the helicity and spin current (Phys. Rev. B 94, 125143 (2016), Appl. Phys. Lett.115, 012401 (2019)). The junctions with ferromagnet and spin Hall material (e.g., Pt) seem useful for the zero-field detection of spin helicity. The spin polarized current and thin film sample are also useful to achieve another important functionality of instantaneous helicity switching. To address these points, we have added the descriptions to the “Conclusion and perspective” section.

Comment(2):

-The term helical is used in a confusing fashion. The helical spin structure means that the spins rotate in the plane normal to the magnetic modulation vector. The general term is magnetic spiral, with the helical and cycloidal structures being the limiting cases. The former is analogous to a structural helix, not the latter. I would suggest to use spiral for the general case and limit the usage of helical for the screw-like structure.

Response(2):

Following the recommendation, we use “spiral” for the general cases and “helical” for the limited cases in the revised manuscript.

Comment(3):

-In Fig. 1b the schematic representation of the mirror plane is misleading: For the case of the magnetic field the mirror plane should be perpendicular to the field. Basically, those states are degenerate, which are connected by the symmetries of the fields. For the current these are mirror planes parallel to the current but for the magnetic field it is only the mirror plane perpendicular to the field. This may be better visualized in the figure.

Response(3):

We did not intend the dotted line in Fig. 1b to be a mirror plane. The similarity between Figs 1 a and b seems to mislead. Figure 1b merely illustrates the chirality dependent effects of electric current and magnetic field on the helimagnetic structure. To avoid this confusion, we have modified the Fig. 1 b.

Comment(4):

In the main text they have the corresponding statement: "The net magnetization direction of the conical spin structure depends on the sign of the product of spin helicity and electric current". Either a reference is needed or this should be explained based on the modified Fig. 1b or in analogy with a simple solenoid.

Response(4):

Actually, we cited a theoretical paper (ref.13) regarding the electric current effect on the helical magnet. While the reference was cited above, our discussion was still based on the reference in the sentence of "The net magnetization direction of....". To discuss this point more clearly, we describe the detail microscopic mechanism of electric current. There are two types of spin-transfer torque induced by an electrical current, as shown in the new Eq. (1). The first term is adiabatic spin-transfer torque, and the second term is so-called β -term originates from the damping and the non-adiabaticity. The beta term produces the conical structure. According to this equation, you can understand why the net magnetization direction of the conical spin structure depends on the sign of the product of spin helicity and electric current. To address these points, we have added the description at 4 page 7th line.

Comment(5):

-The Authors should stress more that Eq. 1 is about a non-reciprocal term to the non-linear resistivity.

Response(5):

The observed 2nd harmonic resistivity has two components. One component is unchanged by the magnetic field reversal, but the other shows a sign change. As the referee suggested, the Eq. 1 describes the latter one, which is caused by the nonreciprocity due to the breaking of both time reversal and chiral symmetries. Following the recommendation, we have added a sentence at 8 page 6th line to emphasize this point.

Comment(6):

-It would be very interesting to see if the chiral domains can also be controlled in situ by current pulses applied in the chiral state and not only by poling through the magnetic transition. Have the Authors tried something along this line?

Response(6):

We also think that this topic is very interesting and directly connected to the application. In principle, the large enough electric current should reverse the chiral domain even deep inside the helical (conical) magnetic state. Nevertheless, we have not observed the chiral domain change in the temperature and magnetic field region far from phase transitions, so far. Besides, a large electric current sometimes broke the micro-fabricated device. We will pursue this issue after fabricating a much thinner single crystal or a thin film. To address this point, we added the description in the “Conclusion and perspective” section.

Comment(7):

-What was the frequency range covered in the present study? Was there any frequency dependence of the effect observed?

Response(7):

To respond to the comment, we newly measured the frequency dependence. Section S4 in the supplemental information shows the result. While the magnetic field independent offset shows some frequency dependence, the intrinsic odd-function component shows little frequency variation.

Comment(8):

-Phrasing and readability can generally be improved throughout the text.

Response(8):

We have reconsidered the whole description in the manuscript and further improved the manuscript with the help of the professional English editor.

Authors' response to the referee 3's comments

Followings are our straightforward responses to responses.

Comment(1):

I do not recommend this paper for publication. The decision is mostly based on the language and on the unclear explanations used in the manuscript. Both points make it very hard to understand the text and to make a decision on the validity/novelty of the

experiments or if it is of interest to the community.

The manuscript needs to be rewritten before being considered to be published anywhere.

Comment(2):

Very bad English: “ To elucidate this issue, a theory paper regarding the spin dynamics in longitudinal helical magnets under electric current seems quite helpful.” “Since the sense of rotation, which is denoted as helicity, is reversed by any mirror operation, it is corresponding to the chirality.”

Comment(3):

Wrong use of articles throughout the whole manuscript.

Response(1),(2),(3)

We have reconsidered the whole description in the manuscript, and further improved the manuscript with the help of the professional English editor.

Comment(4):

The part of controlling the helicity of the spin spiral is not well explained. I think it is the poling process that induces a certain type of spin spiral, which they then detect by eMChE. However, due to bad explanations the reader cannot follow their logical process.

Comment(5):

The eMChE method has been explained in other papers, however, a basic introduction to the main points are important to understand the results. This is the only explanation the reader gets: “While similar 2nd harmonic resistivities are also observed for non-chiral noncentrosymmetric materials, an important feature of electrical magnetochiral effect is that the nonlinear transport is emergent in parallel or antiparallel with the applied magnetic field. This phenomenon has been observed in several chiral helimagnets. Since it has been well established that ρ depends on the chirality, it is useful for the probe.”

Response(4),(5)

To respond to these comments, we have added the descriptions about the mechanism of helicity control, the poling procedure, and the magnetochiral effect. Regarding the

mechanism of helicity control, we have added the equation for the spin-transfer torque (new Eq. 1) and explain how the electric current induces the conical spin structure. The poling procedure has been originally performed for ferroelectrics. We have newly described the background and the effect of poling procedure. As for the magnetochiral effect, we have added the explanation of optical and electrical magnetochiral effects, following the comment.

Comment(6):

The whole explanation about the mirror planes is very poorly and unclearly explained. Figure 2 in the supplementary makes it even worse (what are x,y,z planes? It could be the planes where x,y,z are perpendicular to it, but this does not fit to the sketches.). Why is this even important for the paper? The paper does not discuss the transverse helical structure at all still it is extensively explained in the introduction.

Response(6):

We believe that the mirror symmetry is important to demonstrate the chirality of helimagnet and the discrimination from the multiferroic systems. As the reviewer suggested, x, y, z planes are the planes perpendicular to x, y, z, axes, respectively. The definition was described in the caption of Fig. S2 of the previous version. The confusion seems to be partly caused by the unfamiliarity with mirror operations on axial vectors. The magnetic moments are axial vectors, of which the effects of point group symmetrical operations are different from those of usual (polar) vectors. The sign of spin moment is unchanged by the mirror perpendicular to the spin but reversed by the parallel mirrors. To emphasize this point, we have added the explanation at 4 page 1st line. In addition, we have divided the Figure S2 into two and modified them to avoid the further confusion.

Comment(7):

Fig. 2c How is the black line between the FM1 and helical region (not the dotted line) determined? There are no experimental data points (red). Why does the phase diagram only go up to 100 K even though the text says that the magnetic phase starts at 290 K?

Response(7):

At first, we would like to stress that the precise determination of the phase diagram is not essential in this paper. The magnetic phase diagram is already reported in ref. 23. The purpose of the phase diagram is confirmation of the old result and refinement for

the present sample. We focused the phase diagram below 100 K because the helicity control is performed below 60 K. For this reason, we omitted the description about the transition at 290 K in the revised manuscript. As the referee suggested, the boundary between FM1 and metastable states is ambiguous compared with other transition lines. Judging from the magnetoresistance data in Fig. S4, there is definitely the metastable state at 55 K, maybe at 57 K but it seems to vanish at 60 K. ρ^{2f}_{asym} is more sensitive probe of helimagnetic volume fraction. It is finite at 57 K and disappeared at 59 K. In the revised version, we show the boundary as a thin dotted line to imply the few K ambiguity.

Comment(8):

Fig. 2d why is there no sketch of FM1?

Response(8):

The purpose of this figure is to explain the variation of magnetic structure in magnetic fields along a-axis at 51 K, in which the poling procedure is performed. The most important point is that FAN structure can be viewed as the superposition of two conical structures with different helicities. Therefore we do not show the sketch of FM1. In the revised manuscript, we rather omit the helical spin structure and FM2 structure, and concentrated on the FAN and conical structures in the Fig. 2d.

Comment(9):

What is a field-odd contribution?

Response(9):

The observed 2nd harmonic resistivity has two components. One component is unchanged by the magnetic field reversal but the other shows sign change. The “field-odd contribution” is the latter one. To address this point, we used “odd-function contribution” instead of “field-odd contribution”.

All the changes made are listed below:

1. To respond to the reviewer 2’s comments (1) and (6), we have added the descriptions to the “Conclusion and perspective” section.
2. To respond to the reviewer 2’s comment (2), we use the term “spiral” for the general cases of magnetic structure, and “helical” for the limited cases in the revised manuscript.

3. To respond to the reviewer 2's comment (3), we have modified Fig. 1b.
4. To respond to the reviewer 2's comment (4), we have added the description at 4 page 7th line .
5. To respond to the reviewer 2's comment (5), we have added a sentence at 8 page 6th line.
6. To respond to the reviewer 2's comment (7), we have added the section S4 and Figure S4 to the supplemental information.
7. To respond to the reviewer 2's comment (8) and the reviewer 3's comments (1)(2)(3), we have improve the English and readability throughout the whole manuscript with the help of the Nature research editing service.
8. To respond to the reviewer 3's comments (4)(5), we have we have added the equation for the spin-transfer torque (new Eq. 1) and the descriptions regarding the spin-transfer torque, poling procedure, and the magnetochiral effect.
9. To respond to the reviewer 3's comment (6), we have modified Fig S2 in the supplemental information and added the description at 4 page 1st line.
10. To respond to the reviewer 3's comment (7), we have modified Fig. 2c.
11. To respond to the reviewer 3's comment (8), we have modified Fig. 2d.
12. To respond to the reviewer 3's comment (9), we have used "odd-function component" instead of "field-odd component". We have also added the description about the meaning of "field-odd component "at 7 page 29th line.

REVIEWERS' COMMENTS:

Reviewer #2 (Remarks to the Author):

The Authors sufficiently addressed my comments and question. After the revision of the manuscript, the presentation of the results is clear and accessible.

I recommend the present version for publication in Nature Communications.

Istvan Kezsmarki

Reviewer #3 (Remarks to the Author):

The Authors have addressed and implemented all the questions and the physics and experiments seem scientifically valid. However, the level of English is still very poor and some of the explanations are still missing.

REVIEWERS' COMMENTS:

Reviewer #2 (Remarks to the Author):

The Authors sufficiently addressed my comments and question. After the revision of the manuscript, the presentation of the results is clear and accessible.

I recommend the present version for publication in Nature Communications.

Istvan Kezsmarki

Reviewer #3 (Remarks to the Author):

The Authors have addressed and implemented all the questions and the physics and experiments seem scientifically valid. However, the level of English is still very poor and some of the explanations are still missing.

AUTHORS' RESPONSE:

In the last review process, we reconsidered the whole description in the manuscript, and further improved the manuscript with the help of the professional English editor. Thus, we already did what we can do. We believe that the presentation of the results is now clear and accessible as the reviewer #2 suggested.